# Breaking the cycle: Systematic review of perinatal interventions for parents at risk of child removal

Esther Ariyo[1]*, Victoria Awortwe[2], Ebenezer Cudjoe[3]

1 School of Health and Social Care, University of Essex, Colchester, United Kingdom, 2 Complex Intervention Research in Health and Care, Department of Women's and Children's Health, Uppsala University, Uppsala, Sweden, 3 Department of Psychosocial and Psychoanalytic Studies, University of Essex, Colchester, United Kingdom

* esther.ariyo@essex.ac.uk

## Abstract

This systematic review examined the effectiveness of perinatal interventions aimed at preventing infant removals, with attention to service features, implementation barriers, and enablers. We searched six electronic databases and 15 relevant websites for peer reviewed studies published between 2014 and 2024. Eligible studies evaluated interventions targeting pregnant parents at risk of having another child removed and reported on infant removal outcomes. Independent reviewers screened studies using Covidence. A total of 256 records were obtained, of which six peer reviewed studies covering eight interventions, involving 3,254 pregnant women and 20 professionals met the inclusion criteria. Three studies included comparison groups, including only one randomized controlled trial. Five studies assessed program-level interventions, and one study evaluated a policy change. Risk of bias was assessed using the Mixed Methods Appraisal Tool (MMAT). Two of the three comparative studies indicated that targeted interventions may help reduce infant removals. Four of the six studies highlighted that trauma-informed, relationship-based, and multidisciplinary approaches delivered during pregnancy were associated with reductions in infant removals and improvements in maternal wellbeing, housing stability, substance use, and service engagement. Facilitators of successful implementation included continuity of care, culturally safe and non-judgmental support, and flexible services tailored to family needs. Common barriers were late referrals, limited intervention timelines, mistrust of services particularly among families with prior removals and insecure funding that constrained scale and sustainability. Despite generally positive outcomes, the evidence base remains weak due to small samples, limited diversity, lack of comparison groups, and short follow-up periods. This first systematic review of perinatal interventions for preventing infant removals highlights the need for long term, inclusive, comparative research. It underscores the importance of embedding

**Data availability statement:** All relevant data are within the manuscript and its Supporting Information files.

**Funding:** The author(s) received no specific funding for this work.

**Competing interests:** The authors have declared that no competing interests exist.

early, holistic support in routine services and offers valuable insights for policy and practice on supporting parents with complex needs within the child protection system.

---

## 1. Introduction

Emerging evidence shows a significant portion of child removals involve infants [1]. In Australia, infants (children under one year old) represent the largest proportion of child removals, with a removal rate of 6.5 percent per 1,000 which is significantly higher than the rate of 2.0 per 1,000 observed in other age groups between 2023 and 2024 [2]. Similarly in England, 70 per 10,000 children under one year old were in care proceeding in comparison to 19 per 10,000 children aged between one and four years old [3]. "Care proceedings" provide child protective services like local authorities, or family courts the statutory power to remove a child (<18 years old) from their biological parents where there are concerns for child welfare [4].

Research shows that recurrent care proceedings account for a significant portion of infant removals across countries, with first-time mothers in England, Indigenous groups in Australia, and Black families in the United States of America (USA) identified as particularly affected populations [5–7]. Recurrent care proceedings include "a cycle of returning to court and having subsequent children removed" [8]. An Australian cohort study of 1,629 mothers (2000–2018) found that 20.3% of biological parents who had an infant removed by court order experienced repeat removals, with Indigenous groups representing the largest proportion in national infant removal statistics [9,10]. In Scotland, Wales and England respectively, at least twenty percent of mothers who experience infant removal were first time mothers [11,12]. In England, about 25% of mothers with a prior court-ordered infant removal subsequently faced repeat care proceeding within 10 years [13,14]. In England and Wales respectively, half of mothers involved in care proceedings were subject to repeat care proceedings within eight years and five years respectively [15]. Ninety two percent of Scottish mothers with older children in care proceedings had previously had at least one child removed [16]. Studies in the USA show that birth mothers with a previous child removed have significantly higher odds of experiencing another removal [17], with Black children over represented at 20% in 2021 [18].

Infant removals, though intended to serve the best interests of the child, often lead to significant adversity for parents, creating a cycle of tragic consequences for the families with implications for the society at large [19]. Research suggests that predisposing issues may persist, become compounded, and result in new challenges for the family [10,12]. Parents enter a cycle of addiction and instability following infant removal and these emotional struggles contribute to further involvement with child protection services, recurrent care proceedings, and a reduced likelihood of reunification [20–22]. Mothers who experience infant removal often suffer from emotional and psychological distress [10,20,21]. The removal of an infant at birth can be profoundly

traumatic, negatively affecting women's self-esteem, with many mothers experiencing enduring feelings of inadequacy and shame related to their perceived inability to fulfil their maternal role.

The socioeconomic consequences of infant removal are also severe, with many parents facing reduced welfare entitlements and financial hardship, which complicates their ability to meet reunification requirements [23,24]. Economic struggles combined with emotional distress often result in long-term challenges, including reliance on public assistance programs and increased involvement with services such as housing support, mental health care, and substance abuse programs [20,25]. There are also significant financial implications for society. The costs associated with child protection services, foster care placements, legal proceedings, and social welfare programs place a substantial burden on public resources, leading to a cycle of intergenerational burden on public resources [26,27].

Consequently, in recent times a variety of services, policies, and interventions have been put in place to reduce infant removals and support at risk parents [14]. The services often includes post proceeding support, prenatal support, and targeted support [28]. For example, in the United States, federal changes to the Child Abuse Prevention and Treatment Act now require child welfare agencies to receive notifications about prenatally substance-exposed infants and provide support through Plans of Safe Care programs [29]. Similarly, in Australia and across Europe, policies have been introduced to enhance prenatal reporting and offer early support, preventing infants from entering out-of-home care while identifying situations where removal may still be necessary [30]. In England about 73 services have been locally developed to cater to the unique needs of these families, providing targeted support, creating opportunities for systemic change, often aimed at breaking the cycle of repeat removals through a combination of practical support, emotional care, and legal guidance [28]. The overarching principle of these programs is to offer early, tailored support to parents, thus ensuring that interventions are not reactive but preventive, with the goal of promoting stable family environments and reducing the likelihood of future removals [30].

There is growing but still limited evidence that intensive, relational, and tailored interventions, that combine practical, therapeutic, emotional support and advocacy, with a focus on reducing parental stress and drawing on parents' commitment to their children can be effective in reducing the rate of infant removal [31–33]. However, the overall impact remains inconclusive, as there has been no significant decline in cases of infant removal and may, in fact, be increasing [14]. In England, the likelihood of care proceeding for newborns doubled between 2008 and 2016 [3]. In Australia, rates of infant entry to care increased by 2% per year between 2012 and 2019 [34]. In New Zealand, there was 33% increase in the rate of infants entering out-of-home care within 3 months of birth between 2015–2018 [35]. This highlights persistent gaps in effective service provision, with the specific mechanisms driving positive outcomes still poorly understood.

For example, a recent scoping review raises questions about when support should be offered to families [32]. One major challenge in designing effective interventions is determining the optimal time to initiate support services [36,37]. While some studies argue that interventions should begin as early as pregnancy to achieve the best outcomes [31,38,39], there is no consensus on the most effective timing. Research suggests that pregnancy represents a critical period when expectant mothers are more receptive to making positive life changes, often motivated by their desire to keep their newborns in their care [40–42]. Practitioners also emphasize that early intervention during pregnancy allows necessary assessments and support structures to be put in place before birth [43]. However, there is limited research to substantiate this theory [44]. Therefore, it is important to identify features of interventions that are more likely to achieve positive outcomes for parents and newborns to develop these areas further.

## 1.1. Objective

This systematic review aims to investigate the effectiveness of perinatal interventions designed to prevent infant removals and recurrent care proceedings. Infant removal interventions are initiated either during pregnancy or following the removal of an infant. As our primary outcome of interest is the prevention of infant removal, this review specifically focused on interventions that commence during pregnancy. This approach provide a clear framework for evaluating preventive strategies at a critical juncture before a removal occurs when parents may be most receptive to support. Second, it facilitates a more precise understanding of

the impact of early intervention, which may enhance long-term child welfare outcomes. In addition, pregnancy is a significant point in which service effectiveness is not associated with contraceptives use, implying that parents are supported to develop parenting capacity which are likely sustainable in future [31]. Lastly, by isolating interventions that begin during pregnancy, this review contributes to policy and practice by offering evidence based recommendations on the optimal timing and components of effective prevention programs, ultimately supporting better outcomes for both parents and children. To the best of our knowledge, this is the first systematic review to examine the effectiveness of perinatal interventions aimed at preventing infant removal during pregnancy.

The specific research questions for this systematic review are:

1. What interventions/policies reduce infant removals and recurrent care proceedings for pregnant families?

2. Are interventions particularly effective with different groups of parents (e.g., disabled or ethnic minority parents)?

3. What common elements are shared by effective interventions?

4. What are the enablers and barriers to the successful implementation of interventions for parents?

5. What are the perspectives of parents on the acceptability and usefulness of different interventions?

## 2. Methodology

We conducted this systematic review to investigate the effectiveness of perinatal interventions designed to prevent infant removals and recurrent care proceedings, including the context in which they work, focusing on interventions that begin during pregnancy. We report our results following the Preferred Reporting Items for Systematic Reviews and Meta-Analysis (PRISMA) guidelines [46]. For checklist see S1_Prisma checklist in S1 File. The review protocol was registered in the Prospective Register of Systematic Reviews (PROSPERO), registration number CRD42024561692.

### 2.1. Search strategy

We systematically searched six electronic databases (i.e., APA PsycINFO, PsycARTICLES, Scopus, PubMed, CINAHL, and Web of Science) and hand searched relevant websites (see S2 Appendix A for details). The following terms were used for the search: "Baby Remov*" OR "Infant Entry" OR "Child Remov*" OR "Infant Remov*" OR "Remov* At Birth" OR "Recurrent Care" OR "Repeat Care Proceedings" AND "Intervention" OR "Prevent*" OR "Systemic Solution" OR "Reduc*" OR "Servic*" AND "Parent*" OR "Famil*" OR "Birth Mother" OR "Mother" OR "Father" OR "Maternal" OR "Paternal" AND "Family Court" OR "Child Protective Service*" OR "Care Order" OR "Family Justice" OR "Local Authority" OR "Care Proceeding" OR "Child Welfare Case*" OR "Child Protection". See S2 _Search string for full search strategy used. The snowball method (i.e., references from other identified literature) was also used to identify other relevant studies that were not found in the databases and websites. The search was conducted in July 2024 and rechecked during manuscript preparation. These methods yielded 256 articles for potential inclusion. The study selection process is depicted in a PRISMA flow diagram below in Fig 1.

Studies were included if:

i. It includes an intervention aimed at preventing child removal for pregnant mothers at risk of child removal. The rationale for this exclusion is that assessing the effectiveness of interventions in preventing infant removal becomes significantly more complex when the intervention is implemented at a time when the mother is not currently pregnant. By focusing on interventions that begin during pregnancy, this review ensures that the findings directly address the objective of preventing infant removals.

ii. The study reports on the child protection outcome of the intervention for the baby.

iii. The study was conducted in settings comparable to the UK, where rates of infant removal and repeat removals are high. To ensure relevance, we included research from countries with similar child protection systems (England,

 

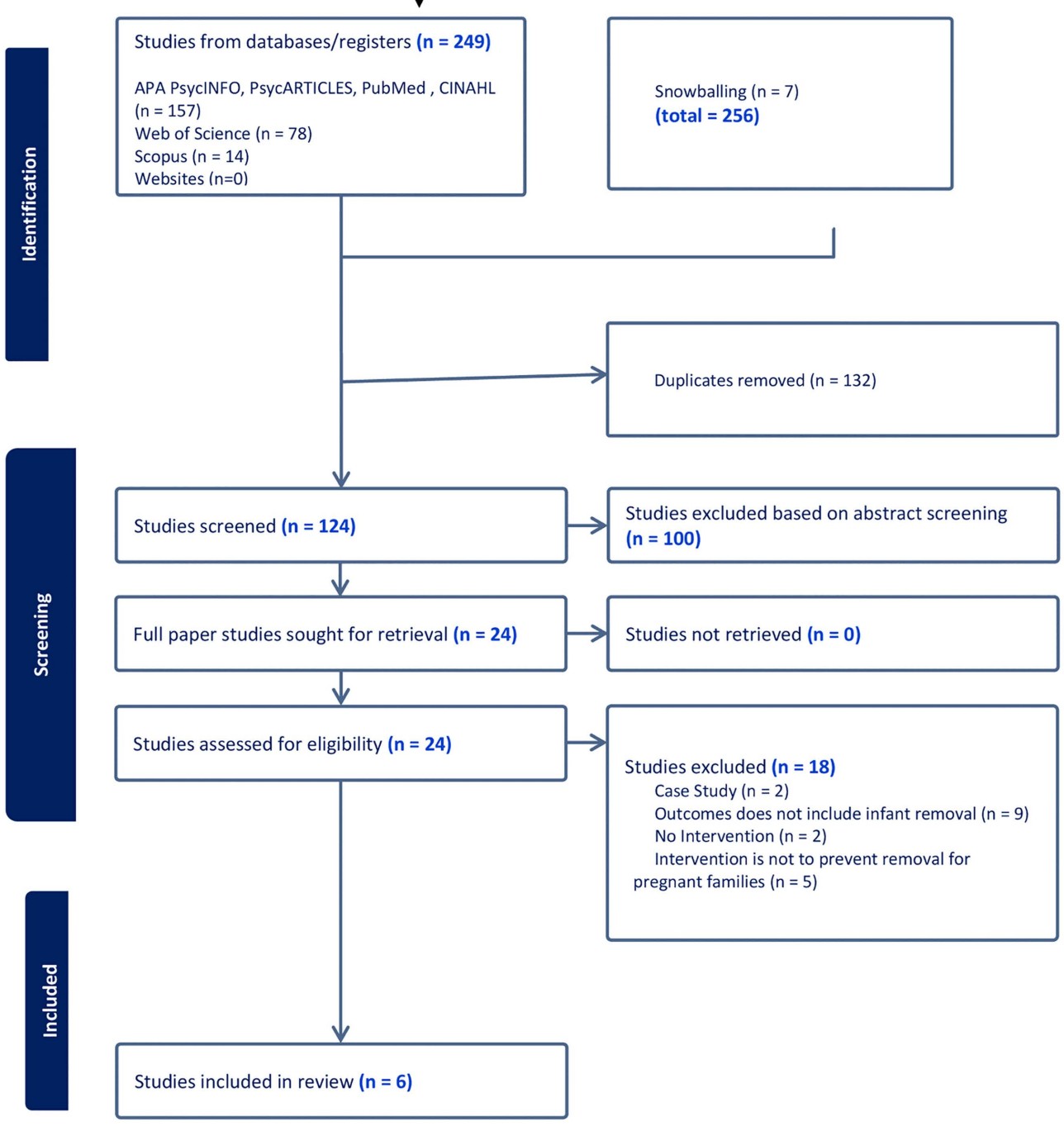

**Fig 1. PRISMA diagram of the study.**

Scotland, Wales, North Ireland, USA, Canada, Australia, New Zealand, Finland, Norway, and Ireland). These countries share a child protection orientation that is broadly comparable to the UK model, in which the state holds statutory powers to intervene in cases of child maltreatment, including the authority to remove children from parental care through

court proceedings. These systems are primarily child-focused, operate under a legal framework that prioritises the child's safety and welfare, and allow for both voluntary and involuntary out-of-home placements. They also share features such as adversarial or quasi-adversarial court processes, mandatory reporting duties, and a focus on preventing re-entry into care, making them relevant for comparison with the UK context of high rates of infant removal and repeat removals.

iv. The study is peer reviewed.

v. The study is published in English. This language restriction was applied for two main reasons: first, English is the primary language of publication in the countries where research on recurrent child removal has been most prominent; and second, it ensured consistency in data extraction and quality appraisal, minimizing the risk of misinterpretation due to translation.

vi. The paper is published between 2014 and 2024. The date allows for recent overview of literature, and it aligns with the timing on when the research on recurrent infant removal began to emerge [7].

The eligibility criteria were designed to ensure that the review findings would be relevant for policymaking and capable of informing practice. Studies were excluded if they;

(i)   described an intervention without providing an evaluation of its outcome.

(ii)  were interventions focused only on support available to birth parents after child removal.

(iii) described reunification of families (i.e., reuniting families after care proceedings have closed).

(iv) described intervention for adoptive parents

(v)  described the involvement of child protective services (CPS) without mention of child placement.

(vi) were abstracts, conference proceedings, reviews and grey literature.

Data screening and data extraction was done in Covidence. Titles and abstract screening and full text screening were conducted independently by two reviewers (EA & VA). Interrater agreement during screening was moderate, with a proportionate agreement of 71% and a Cohen's kappa of 0.39, indicating fair agreement [45]. Discrepancies were resolved through discussion to reach consensus. Data extraction was completed by all authors and consensus was reached by two authors for each included study. We extracted detailed information from each study, including the author, sample size, population sample, country of study, dataset used, comparison groups, study aim, key findings, data source, study design, other measured child and parental outcomes, intervention description, implementation and delivery personnel, recruitment process, intervention timeframe, acceptability, barriers and enablers.

## 2.2. Data synthesis

For research question one, which focused on identifying interventions and policies that reduce infant removals, we used descriptive analysis to summarise and categorise the types of interventions reported across studies. This approach allowed us to map the range, and characteristics of interventions in a structured way. For research questions two to five, which explored effectiveness across different parent groups, common elements of interventions, implementation factors, and parents' perspectives, we used thematic analysis [46]. This approach enabled us to synthesise qualitative and mixed-methods findings, identify recurring patterns, and generate overarching themes across studies. We then provide a narrative of all findings in alignment with our study objectives. Data was synthesised per each research objective. We did not conduct a meta-analysis given the substantial heterogeneity in the results of individual studies which could undermine the validity of the pooled results [36].

## 2.3. Quality assessment approaches

The risk of bias of the included studies was assessed using the Mixed Methods Appraisal Tool (MMAT) approach [47]. Each study was evaluated against the five criteria relevant to its study design (quantitative or mixed methods). Two researchers independently conducted the appraisal, and any disagreements were resolved through discussion. The quality of evidence for quantitative outcomes (infant removal, child protection service involvement, and maternal mental health) was assessed using the Grading of Recommendations, Assessment, Development, and Evaluation (GRADE) guidelines [48]. This approach considers five domains: risk of bias, inconsistency, indirectness, imprecision, and publication bias. For qualitative outcomes (acceptability of services, barriers, and enablers to service implementation), the GRADE-CERQual framework [49] was applied, which evaluates four domains: methodological limitations, relevance, coherence, and adequacy of data.

Both GRADE and GRADE-CERQual provide a structured process for determining the certainty of evidence by examining factors that may increase or decrease confidence in the findings. All researchers discussed and reached consensus on the ratings for each outcome. The certainty of evidence across the reviewed outcomes varied.

## 3. Results and discussions

### 3.1. Description of included studies

**3.1.1. Study characteristics.** This review included six empirical peer-reviewed studies evaluating eight perinatal interventions. Five studies examined direct parental support programs, and one focused on policy reform [30]. Studies were conducted in Australia [30,50,51] and the United Kingdom [31,52,53], comprising four quantitative and two mixed-methods designs. Methodologies includes cohort studies [50,51], post-program evaluations [53], cross-sectional descriptive study [30], and mixed-methods evaluations [31,52]. Three studies included a comparison group of women with eligibility criteria similar to the intervention group [50–52].

**3.1.2. Participants.** The studies collectively included 3,254 pregnant mothers and 20 professionals. The sample sizes ranged from 53 [53] to 1,988 [51]. None of the included studies reported on any intervention with fathers or engagement with fathers. The pregnant mothers in these studies were predominantly high-risk individuals facing multiple vulnerabilities. Common characteristics included young maternal age, low socioeconomic status, housing instability, previous child removals, experiences of domestic violence, substance misuse, and mental health challenges. All studies included mothers that were at risk of child removal based on previous child removal [53,54], or referral to child protection service [30,52], vulnerable Indigenous (First Nations) mothers [50,51].

Data sources included program data collected during the implementation of the intervention [50,53], administrative datasets [30,51] and a combination of program data and validated measures [31,52]. Recruitment occurred mainly through healthcare and local authority referrals, although self-referral was possible in one study [50].

**3.1.3. Interventions.** The interventions were implemented by government and community agencies [30,31,51,53], non-profit organizations [31], and academic institutions [52]. Two studies reported that interventions were delivered by a team of nurses/midwives and community workers [50,51], while three studies report that a mix of social workers, family support, and professionals delivered the intervention [31,52,53]. The prenatal reporting policy was managed by a prenatal support team within the child protection unit in Australia Capital Territory [30].

All five program-based studies reported that interventions incorporated individualized case planning, with professionals working closely with parents to address challenges and build parenting capacity, in order to reduce the risk of infant removal. Core components included therapeutic support, counselling, parenting training, and referrals to medical and community services. The "Parents Under Pressure" (PuP) programme further includes trainings on emotional regulation, problem solving and stress management [52]. The Early "Family Drug and Alcohol court" (FDAC) intervention also includes a judicial oversight and drug testing along professional support to families [53]. The prenatal reporting policy

includes the notification of child protection services during pregnancy about an unborn child that is considered at risk, with the aim of preventing harm after birth or having the baby taken into care and it offered voluntary assessments, child protection assessment at birth, referral to support services, and case work support for the coordination of services [30].

**3.1.4. Interventions duration.** Four studies report that interventions were planned to start in the second trimester. Two interventions initiated support between 13 and 18 weeks [50,52], and prenatal reporting was majorly in the second or third trimester [30]. Shaw [53] reported late start in the third trimester due to late referral although program was planned to start with women at earlier gestation age. Cox, McPherson [31] and O'Dea, Roe [51] noted pregnancy based intervention but did not specify gestational age. Interventions lasted up to six weeks post birth to five years post birth. Three interventions provided support up untill one year after birth [31,52]. Shaw [53] reports up to 2 years post-natal support, while O'Dea, Roe [51] reports six weeks post-birth support. Cox, McPherson [31] also reported an intervention that lasted for 5 years post birth based on parents' availability.

**3.1.5. Outcomes reported.** Outcomes reported included child removal post birth, child protection service involvement status, parental psychosocial wellbeing, parental social support, housing stability, changes to economic and financial situation. We synthesised evidence for research questions two to five.

### 3.2. Interventions that reduce infant removals and recurrent care proceedings for pregnant families

**3.2.1. Infant removal.** Of the five program-based intervention studies, four studies suggest some evidence of reduction in child removals. O'Dea, Roe [51] reports a statistically significant reduction difference between groups, with 0.95% of infant removal from 944 women in the intervention group compared to 2.97% in the control group consisting of 1044 women that received standard routine care after controlling for risk variables. Segal, Nguyen [50] also reports that child removal in the intervention group of 256 women was 7% compared to 9% in the control group of 563 women within infants first 5 years. The control group were unenrolled mothers with the same eligibility criteria as mothers enrolled in the intervention program. Cox, McPherson [31] and Shaw [53] had no comparison group, however, the authors suggest that the result of the intervention may signify evidence of reduction in child removal. In Cox, McPherson [31] 96 out of 127 infants remained with their family members across the three interventions evaluated. Despite challenges of engagement reported with Early FDAC intervention, the study reported that 18 infants out of 28 remained with their families [53].

However, Harnett, Barlow [52] found that women in PuP control group (routine care) had no child removal, whereas 26% of children in the intervention group experienced legal proceedings and removal. The authors suggests that this likely reflects a more timely and informed decision making by practitioners, and action to prevent longer term harm. Similarly, the Parental reporting policy did not show evidence of success in reducing child removal. Twelve percent of infants reported were removed, and this outcome is similar to those of substance-using women in Australia [30].

**3.2.2. Other child protection outcomes.** Three studies reported additional child protection outcomes relating to engagement with child protection services or agencies. Cox, McPherson [31] reported that 12 children who had been removed from their mothers before engagement with the intervention were successfully returned. Segal, Nguyen [50] found that children of younger mothers in the intervention group had significantly lower rates of involvement with child protection services and spent fewer days in care. Harnett, Barlow [52] assessed child protection status and reported that safeguarding status improved after 12 months for 42% of those in the intervention group compared to 14% in the routine care group. The comparison group had a higher proportion of cases where safeguarding status remained unchanged. Further details of the findings are in Table 1.

**3.2.3. Parental outcomes.** Parental outcomes were assessed in two studies, focusing on psychosocial well-being and social support [31,52]. This was grouped as maternal psychosocial wellbeing and maternal social wellbeing (includes housing stability, better access to services, and enhanced social support) for quality assessment. Harnett, Barlow [52] reports improvements in psychosocial wellbeing, with reductions in depression, anxiety, and stress, as well as increased social support among parents engaged in the intervention. Cox, McPherson [31] found modest improvements in housing

**Table 1. Summary of included studies.**

| Title | Country | Methodol-ogy | Interven-tion/ Policy name | Sample | Child removal outcome | Other outcomes reported |
|---|---|---|---|---|---|---|
| Child protection outcomes of the Australian Nurse Family Partnership Program for Aboriginal infants and their mothers in Central Australia [50] | Australia | Comparative Cohort study | Nurse Family Partnership Program (FPP) | FPP = 291; Con-trol = 563 mothers | FPP group had 7% placement in care within the first five years compared to 9% in the control group | Lower rates of involvement with child protection services for mothers less than 20 years in the FPP group when compared to the control group |
| A Proof-of-Concept Pilot for an Intervention with Pregnant Mothers Who Have Had Children Removed by the State: The Early Family Drug and Alcohol Court Model [53] | England | Quantitative program pilot review | Early FDAC | 53 families | 36% (10 out of 28) of infants were removed from their parents. There was no com-parison group | None |
| Assessing capacity to change in high-risk pregnant women: a pilot study [52] | United Kingdom | Mixed meth-ods quasi-experimental study | Parents under Pres-sure (Pup) programme | PuP = 31; Control = 29 women, 20 professionals | There was 26% removal order for birth parents in the PuP group while there was none in the control group at 12 months | Safeguarding status improved by 42% in the PuP group compared to 14% in the control group; There was no changes in safeguard-ing status for 10% in PuP group compared to 40% in control group; There was significant decrease in depression, anxiety, and stress and increase in social support at $P < 0.01$ for the PuP group |
| Prenatal reporting to child protec-tion: characteristics and service responses in one Australian jurisdic-tion [30] | Australia | Quantitative analysis of admin-istrative database | Prenatal reporting policy | 117 women | 12% of infants reported were removed within their first 100 days. The result is similar to that of substance using women. | None |
| Breaking the cycle: Effect of a multi-agency maternity service redesign on reducing the over-representation of Aboriginal and Torres Strait Islander newborns in out-of-home care: A prospective, non-randomised, intervention study in urban Australia [51] | Australia | Non-randomised interven-tional study | Birthing in our Community | BioC = 944; Con-trol = 1044 women | 0.95% (9 women out of 944) in the BioC group had their infants removed in comparison to 2.97% (31 women out of 1044) in the control group | None |
| Reducing recurrent care proceed-ings: Building a local evidence base in England [31] | England | Mixed methods | Names were not provided for the three services for confidential-ity reasons | 182 women | 75.6% (96 out of 127) women had their children with them or family | 12 children taken previously before engagement returned to back parents; There was modest improvements in housing stability, reduced substance use, closure to use of probation service and a decline in partner abuse. For psy-chosocial wellbeing most did not improve and some deteriorated |

stability, reduced substance use, closure to use of probation service and a decline in partner abuse. However, there were mixed reports for mental health outcomes, with some participants requiring increased medication, and others experiencing deterioration in psychological well-being or experiencing no changes. Economic outcomes also varied, as employment increased slightly in two interventions while unemployment rose in one. Overall, both studies showed improvements in

social stability but mixed results in economic and mental health outcomes. Table 2 presents a summary of intervention outcomes reported across studies.

### 3.3. Are interventions particularly effective with different groups of Parents (e.g., disabled or minority parent, age, or ethnicity)?

While the evidence in this review is limited, two studies [50,51] reports that interventions were specifically designed for Indigenous groups. Both reports reduced infant removal among Indigenous mothers. Other studies did not report on ethnicity [30,53] or had majority of intervention participants as British white [31,52].

Furthermore, the age statistics within the studies indicate that interventions could be more relevant for younger mothers, as most studies reviewed involved participants with a median or average age between 24 and below 30 years. For instance, Segal, Nguyen [50] reported a median age of 24.5 years in the intervention group, with a comparison group average of 25.6 years. They reported better child protection outcomes for mothers less than twenty years. Shaw [53] noted that parents receiving the intervention were more likely to be in their twenties. Harnett, Barlow [52] recorded a mean participant age of 24 years, and Taplin [30] also reported a median age of 24.5. Although O'Dea, Roe [51] did not provide exact age statistics, the study included teenagers. Similarly, Cox, McPherson [31] described the majority (53%) of participants as being between 20 and 29 years old. While direct comparative analysis by age is limited, these consistent age patterns across studies suggest that younger mothers are a primary population for these interventions and may be especially responsive to support during the perinatal period. However, there remains a lack of focused research on disabled parents and ethnic minorities outside Indigenous populations. Table 3 includes summary of findings for research question two across included studies.

**Table 2. RQ1: Child removal, other child protection outcomes, and parental outcomes.**

| Theme | Infant Removal | Other Child Protection Outcomes | Parental Outcomes |
|---|---|---|---|
| O'Dea, Roe [51] | ✓ | Not reported | Not reported |
| Segal, Nguyen [50] | ✓ | ✓ | |
| Cox, McPherson [31] | ✓ | ✓ | ✓ |
| Harnett, Barlow [52] | ✗ (higher removals) | ✓ | ✓ |
| Shaw [53] | ✓ | Not reported | Not reported |
| Taplin [30] | ✗ (no reduction) | Not reported | Not reported |

✓ intervention reports reduction for infant removal rate.

✗ intervention reports higher or no difference in removal rate.

**Table 3. Acceptability, age, and ethnicity or group (RQ2 & RQ5).**

| Authors | Acceptability of Intervention | Age Description | Ethnicity or Group |
|---|---|---|---|
| O'Dea, Roe [51] | Yes | ✓ (teenagers included) | ✓ (Indigenous-focused) |
| Segal, Nguyen [50] | Not reported | ✓ (median age 24.5) | ✓ (Indigenous-focused) |
| Cox, McPherson [31] | Yes | ✓ (53% aged 20–29) | Not reported |
| Harnett, Barlow [52] | Yes | ✓ (mean age 24) | Not reported |
| Shaw [53] | ✓ (low engagement) | ✓ (mostly 20s) | Not reported |
| Taplin [30] | Not reported | ✓ (median 24.5) | Not reported |

### 3.4. Common elements in successful interventions

This section reports all the common elements in all the six intervention that reported reduction in child removal.

**3.4.1. Continuity of care and trusting relationships.** Across all the successful interventions, there was a continuous, relationship-based support by a consistent caregiver over a period of time. In the *Birthing in Our Community* (BiOC) model, each woman was assigned a known midwife throughout pregnancy and postpartum, enabling deep trust and regular engagement [51]. Similarly, the *Family Partnership Program* (FPP) in Central Australia involved nurse home visitors who developed long-term, trusting relationships with Aboriginal mothers [50]. Cox, McPherson [31] reported that the regular and consistent contact provided by key workers was valued by mothers and considered crucial to the service effectiveness. In the Early FDAC, families are supported by a consistent key worker and work with the same judge throughout the proceedings [53]. This aligns with observations in the literature that relationship-based practice and continuity of practitioner contact underpin effective intervention delivery for parents at risk of child removal [32]. Trusting bonds can allow mothers engage honestly with services, increasing the chances of safe parenting and reducing the risk of removal.

**3.4.2. Multidisciplinary wraparound support.** This review finds evidence that holistic services which address multiple risk factors and draw on multi-professional support are more likely to be successful in reducing child removals. The interventions included practical support that addressed social determinants, such as stable housing, financial aid, and mental health services. The BiOC model co-located midwives, Indigenous Family Support Workers, and other health professionals in a community hub, offering services such as legal aid, housing assistance, and child protection advocacy under one roof [51]. The FPP linked primary healthcare, social services, and child protection into one coordinated intervention [50]. Cox, McPherson [31] reported that mothers perceived support as comprehensive because key workers collaborated with other professionals and community services to provide support for the parents. The Early FDAC likewise allowed parents to access a network of community resources along with a therapeutic team consisting of children's social workers, related professionals and individuals with personal experience of substance misuse and/or care proceedings [53]. These models addressed complex social and health issues that often contribute to removals, such as unstable housing and family violence. It is possible that the success of a holistic approach to intervention could be due to the fact that it requires multiple services and professionals capable of meeting the differing needs of mothers and their infants [32].

**3.4.3. Trauma-informed and non-judgmental approach.** Interventions that used trauma informed and non-judgemental approach were successful with the engaged participants. The BiOC team created a non-judgmental space described as "free of judgement and stigma," where mothers could safely access services and share challenges [51]. Cox, McPherson [31] suggested that key workers developed trust with families by acknowledging their past trauma. Mothers perceived the support was non-judgemental and practitioners focused on emotional safety and healing rather than punitive action. Segal, Nguyen [50] also mentioned that there was respectful care through home visits, and recognition of a trauma-informed practice for Indigenous families with intergenerational adversity. In the Early FDAC program, parents were treated with respect and empathy and encouraged to believe that there could be positive change for their family when they commit to the process [53]. Evidently, trust and non-judgmental relationships are foundational for engagements, especially for mothers who have experienced trauma, stigma, and repeated service failure. Studies have consistently shown that parents value emotional safety, honesty, and kindness from practitioners who are persistent yet respectful [55]. This review aligns with broader literature recognizing that families with complex trauma histories require assertive outreach and extended time to build trust [56].

**3.4.4. Strengths-based, person-centered focus.** Effective interventions used a strength enabling approach, empowering parents to care for their children safely. The studies indicates that key workers had the flexibility to provide person-centered support, ensuring that services were based on the needs of the mothers. Cox, McPherson [31] suggested that mothers were empowered to develop competence that will enable them keep their children in the future through personalised support. In BiOC, care teams emphasized mothers' efforts and resilience during case planning with

child protection [51]. Similarly, FPP promoted family preservation by building parenting skills and highlighting strengths [50]. These programs aimed to support parenting capabilities, which contributed to fewer removals. Summary of themes across studies is in Table 4.

### 3.5. Enablers of successful implementation

A key factor contributing to successful intervention implementation was the presence of strong multi-agency partnerships and the involvement of community-driven service delivery. Interventions co-designed with the target community (such as Indigenous-led programs) and delivered via integrated partnerships facilitated engagement [51,53], provided opportunity to provide multidisciplinary (health, social, legal) comprehensive support to families [31,53] and allowed for adaptation of intervention to the cultural context [50]] Cox, McPherson [31] and Shaw [53] reported the use of cross-sector community services to empower families (Shaw, 2021). BiOC's multi-agency partnership model, led by Indigenous organisations, successfully reduced newborn removals through culturally safe, multi sectoral support to women. Key characteristics included Indigenous leadership and governance, continuity of care provided by dedicated midwives and Indigenous family support workers, integration of cultural integrity frameworks into staff training, and the creation of community hubs that fostered connection, peer support, and engagement with indigenous elders [51]. Segal, Nguyen [50] reports the adaption of the Nurse Family Partnership (NFP) to community levels and implementation by an Indigenous community-controlled health service

### 3.6. Barriers to successful implementation

**3.6.1. Late identification and short intervention windows.** Some studies reported that many high-risk pregnancies are only flagged to services late in gestation, leaving little time for meaningful intervention [30,53]. Starting interventions late limits their potential impact. Moreover, supports often end too soon after birth [30,31]. Short-duration involvement may not be sufficient for parents to demonstrate and sustain change, especially when underlying issues (addiction recovery, mental health treatment) require longer-term engagement [53]. Delayed referrals and the short duration of many interventions highlight the need for proactive identification systems and longer-term support to provide families with a meaningful opportunity for change. Although one included intervention allowed for self-referral, uptake was limited likely due to low awareness among prospective service users. To address this, some research have recommended increasing awareness of preventative services to promote timely and appropriate referrals, including the option of self-referral where suitable [57,58]. Notably, interventions offering extended durations of support appeared to have a greater impact on reducing infant removals compared to those with shorter timelines [31].

**3.6.2. Fear of child removal and distrust of services.** Distrust of services especially among families with prior child removals also emerged as a significant barrier, highlighting the need for culturally safe, community led approaches that foster genuine engagement. Families with prior child removals often distrust social workers and interventions,

Table 4. RQ3: Common elements in successful interventions.

| Theme | O'Dea, Roe [51] | Segal, Nguyen [50] | Cox, McPherson [31] | Shaw [53] | Taplin [30] | Harnett, Barlow [52] |
|---|---|---|---|---|---|---|
| Continuity of Care and Trusting Relationships | ✓ | ✓ | ✓ | ✓ | Not reported | ✓ |
| Multidisciplinary Wraparound Support | ✓ | ✓ | ✓ | ✓ | ✓ | Not reported |
| Trauma-Informed and Non-Judgmental Approach | ✓ | ✓ | ✓ | ✓ | Not reported | ✓ |
| Strengths-Based, Person-Centered Focus | ✓ | ✓ | ✓ | ✓ | Not reported | ✓ |

Studies that reports the themes have been marked as ✓.

perceiving them as punitive rather than supportive and programs that fail to establish early trust may struggle with engagement. Taplin [30] reports that distrust of services was a significant barrier to engagement with antenatal care for those at risk of child removal which may have consequently led to missed opportunities to receive support. Shaw [53] suggested that the low uptake of Early FDAC could be due to mistrust among mothers with previous child removal, and lack of local legal advice to take up the program. Harnett, Barlow [52] indicates that women with histories of trauma or previous removals were hesitant to fully engage, and that practitioners in the PuP intervention facilitated participation by establishing trust with the women. The Birthing in Our Community (BiOC) program also addressed institutional distrust through Indigenously-led, community-controlled model of service to facilitate engagement [51]. Cox, McPherson [31] highlighted that many mothers entering the service felt disempowered by past child protection interventions and the services emphasized trust-building as essential strategies for overcoming resistance rooted in stigma and past negative experiences.

**3.6.3. Funding and sustainability.** Two studies highlighted that interventions are often implemented as pilots with insecure or short-term funding, creating challenges for scalability and sustainability [31,53]. Cox, McPherson [31] reports that the three recurrent care services they evaluated were locally developed and financed through a mix of local authority budgets, charitable grants, and short-term commissioning. This uncertainty limited their ability to retain staff, plan strategically, and expand provision. For example, one service relied on charitable trust grants, while another depended on local authority innovation funding. The authors noted that "none of the services were fully secure in their funding beyond the initial contract period," leading to instability in staffing, continuity of care, and delivery.

Similarly, Shaw [53] reported that the Early FDAC pilot, though promising, faced recruitment delays and operational difficulties linked to fixed-term funding. The study emphasised that while such services addressed emerging needs, their lack of integration into mainstream provision left them vulnerable to closure once pilot funding ended. This insecurity constrained their reach and limited investment in staff development, innovation, and wider dissemination.

These examples underscore the structural barrier posed by funding instability. Even where interventions demonstrate positive outcomes for families, their long-term impact depends on secure, sustained investment and integration into established service systems. Without this, innovative services risk remaining small-scale, time-bound, and inaccessible to many of the families who could benefit most.

Ultimately, investing in enablers and anticipating barriers can improve the likelihood that vulnerable parents receive the help they need, thereby better protecting children and strengthening families. The summary of enablers and barriers identified across included studies is in Table 5.

## 3.7. Perspectives of parents on the acceptability and usefulness of interventions

The research question included the acceptability and usefulness of various interventions from parents' perspectives. However, none of the included studies directly examined parents' views on the usefulness or acceptability of the interventions.

**Table 5. RQ4: Enablers and barriers to successful implementation.**

| Theme | O'Dea, Roe [51] | Segal, Nguyen [50] | Cox, McPherson [31] | Shaw [53] | Taplin [30] | Harnett, Barlow [52] |
|---|---|---|---|---|---|---|
| Presence of strong multi-agency partnerships and community-driven service delivery | ✓ | ✓ | ✓ | ✓ | | |
| Late identification and short intervention windows | | | ✓ | ✓ | ✓ | |
| Fear of child removal and distrust | ✓ | | ✓ | ✓ | ✓ | ✓ |
| Funding and sustainability | | | ✓ | ✓ | | |

Studies that reports the theme have been marked as ✓.

Four studies provided indirect insights through discussions of service user engagement, client experience, or service participation [31,51-53]. As only one study focused on a policy-level intervention, data on the acceptability of such interventions were not synthesised [30].Consequently, the synthesis of acceptability data was limited to program-level interventions.

O'Dea, Roe [51] reported that women developed a sense of belonging, attended health facilities earlier and more frequently during pregnancy, and engaged with integrated services including having trusted relationships with practitioners. Harnett, Barlow [52] highlighted that service users had positive experiences and expressed satisfaction with the PuP program. Similarly, Cox, McPherson [31] indicated that service users viewed staff as helpful and supportive.

However, Shaw [53] reported low engagement levels with the Early FDAC intervention in comparison with the FDAC program (a similar program to the Early FDAC). Only 28 out of 53 referrals to the program fully engaged with the program. They suggested that reasons could be due to parents being less convinced of the program's benefits, poor relationship with the professionals that made the referrals, limited awareness by local lawyers and the potential that parents may not have taken pre-proceedings as seriously as formal legal proceedings.

While further research is needed on parental perspectives, for those studies that did address acceptability and engagement of interventions, reports were high (as displayed in Table 3).

### 3.8. Quality of evidence

**3.8.1. Evidence for intervention outcomes.** Using the GRADE approach, the certainty of evidence for infant removal and other child protection outcomes was assessed as low, reflecting methodological limitations and imprecision in outcome definitions. Evidence for parental outcomes was rated as very low, largely due to small sample sizes and limited follow-up periods.

**3.8.2. Evidence for effectiveness with different groups of parents.** The certainty of evidence regarding differential effectiveness across parent groups was assessed as low, reflecting the small number of studies, limited diversity in participant samples, and restricted subgroup analyses.

**3.8.3. Evidence for common elements across studies.** Assessment with GRADE-CERQual indicated moderate confidence in all themes identified. While themes were consistently observed across studies, they were sometimes reported without sufficient depth or specification of underlying mechanisms, thereby limiting explanatory power.

**3.8.4. Evidence for Enablers and Barriers to Effective Interventions.** Confidence in evidence regarding intervention enablers was rated as moderate, though variability in reporting depth and concerns about data adequacy were noted. Evidence for barriers (e.g., late referrals, funding constraints, and service distrust) was assessed as low. These barriers were frequently identified but often discussed only superficially, lacking rich participant perspectives and relying mainly on descriptive, program-level information.

**3.8.5. Evidence for parents' perspectives on acceptability and usefulness of interventions.** Confidence in evidence on service acceptability was rated as high. Multiple data sources provided coherent and consistent findings, indicating that parents perceived the interventions as acceptable, respectful, and beneficial.

Overall, these findings suggest stronger evidence for how services are perceived than for their measurable outcomes or the mechanisms underpinning effectiveness. Details are in Table 6.

### 4. Conclusions and implications

This review synthesised evidence from six empirical studies evaluating eight perinatal interventions aimed at preventing infant removal from parental care. While it shows some promise of positive outcomes, evidence needs to be strengthened.

Four of the included studies suggest evidence of reduction in baby removal. Positive outcomes extended beyond reduction in child removal, including improvements in maternal wellbeing, reduced psychological distress, increased stability in housing and finances [31,32]. Some studies also reported reunification of previously removed children and

**Table 6. Summary of certainty and confidence of evidence.**

| Outcome | Assessment Type | Number of Studies | Certainty/ Confidence | Reasons | Comments |
|---|---|---|---|---|---|
| **Infant Removal** | GRADE (quantitative) | 6 | Low | Some risk of bias (missing comparators, confounding), small sample sizes, limited RCTs, imprecision in outcome interpretation in two studies [31,53] | Despite methodological limitations, consistent findings suggest moderate benefit from early, integrated perinatal interventions. |
| **Other Child protection outcomes** | GRADE (quantitative | 3 | low | Moderate risk of bias (non-randomised designs), inconsistent measurement of "involvement", imprecision in sample sizes and subgroup reporting | Evidence indicates potential reductions in service involvement (e.g., fewer days in care, reunification), but design and definition variations reduce certainty. |
| **Parental Outcomes** | GRADE (quantitative) | 2 | Low | Mixed outcomes across studies, small samples, lack of long-term follow-up | Some evidence of reduced stress and increased social support in one study [52], but insufficient |
| **Effectiveness with Different Parent Groups (e.g., ethnicity, age, disability)** | GRADE-CERQual (qualitative) | 6 (2 focused on Indigenous groups; 4 reported age patterns; 0 on disability) | Low | Few subgroup-focused studies; limited diversity; minimal reporting on disability and non-Indigenous minorities; lack of comparative analyses | Indigenous parents and younger mothers (under 30) may benefit most, but confidence is reduced due to under-reporting and reliance on descriptive data. |
| **Acceptability of Service** | GRADE-CERQual (qualitative) | 4 | High | Minor concerns about sample size; strong methodological clarity, coherence, and data adequacy | Consistent themes of culturally positive client experiences across studies. |
| **Enablers to implementation**[a] | GRADE-CERQual | 4 | Moderate | Variability in reporting depth; moderate concerns about data adequacy | Enablers like continuity of care, cultural safety, and flexible support were consistently reported, though detailed mechanisms varied across studies. |
| **Barriers to implementation**[a] | GRADE-CERQual | 5 | Low | Limited depth on some barriers (e.g., funding); mostly descriptive and program-level reporting | Common barriers include late referrals, funding constraints, and service distrust, often discussed briefly or without deep participant insight. |
| **Common elements of effective interventions**[a] | GRADE-CERQual | 4 | Moderate | Moderate concerns about coherence and data adequacy; themes reported but not always fully explained | Recurring themes include trauma-informed care, relationship-based support, and early pregnancy engagement, supporting moderate confidence. |

[a]: Certainty of evidence was similar across the themes included in the research objective.

decreased involvement with child protection services over time [31,50,52]. These findings underscore the potential of holistic, early support,particularly during pregnancy a period identified by several studies as a window of opportunity when parents may be more receptive to change when delivered with compassion and flexibility in providing for long-term family stability and improved wellbeing for both parents and children [40,41].

Although this review focused on the effectiveness of perinatal interventions aimed at reducing infant removals, it is important to acknowledge critiques that increased surveillance within child protection systems can have harmful consequences, particularly when disproportionately applied to minority and marginalised groups [59,60]. Such practices may reinforce stigma and perpetuate inequities, with Indigenous families and other racial or cultural minorities often facing greater scrutiny and intervention despite underlying systemic disadvantage [61,62]. Addressing these concerns requires culturally safe, community-led services, as well as broader structural reforms, that would explicitly challenge bias and promote fairness in child protection decision-making [9].

Importantly, this review indicates a need for greater quality and strength of evidence on the effectiveness of perinatal intervention for families at risk of infant removal. Although there are evaluation reports that address recurrent care proceedings and infant removal, only six peer-reviewed studies met the inclusion criteria for this review. These studies varied in their design, intervention structure, and outcome measures. Most included studies lacked comparison groups or long-term follow-up data, making it difficult to draw definitive conclusions about the effectiveness and the sustained impact of interventions beyond the early postnatal period. In addition, the review found little evidence regarding parental perspectives, fathers' engagement and the effectiveness of interventions across diverse parent populations, such as disabled parents or ethnic minorities outside Indigenous groups.

This gap in the literature underscores the need for robust, comparative longitudinal research that tracks families over extended periods and evaluates the enduring effects of perinatal support, and especially in the context where rates of recurrent care proceedings still persist [14]. Long-term studies would be particularly valuable to inform intervention designs and implementation. Moreover, research that incorporates the lived experiences and perspectives of parents themselves would help to assess the acceptability and relevance of services; an area notably underexplored in current literature.

Conclusively, this review highlights the potential of perinatal interventions to improve outcomes for families at risk of recurrent infant removal, offering early and holistic support at a critical point in the parenting journey. It is the first systematic reviews to focus exclusively on pregnancy-initiated interventions in this context, providing a foundation for future policy, practice, and research. While the findings are promising, they also expose significant gaps in the evidence base particularly regarding long-term impact, diverse populations, and parent perspectives. Addressing these gaps through inclusive and comparative longitudinal research will be essential to strengthen the design and delivery of effective, equitable interventions for vulnerable families.

## Supporting information

**S1 File. Prisma checklist.**
(DOCX)

**S2 Appendix. Search string.**
(DOCX)

**S3 File. Risk of Bias using MMAT tool.**
(DOCX)

## Acknowledgments

We would like to acknowledge the contributions of the authors of the studies included in this review.

## Author contributions

**Conceptualization:** Esther Ariyo.

**Data curation:** Esther Ariyo, Victoria Awortwe, Ebenezer Cudjoe.

**Formal analysis:** Esther Ariyo, Victoria Awortwe, Ebenezer Cudjoe.

**Investigation:** Esther Ariyo.

**Methodology:** Esther Ariyo.

**Supervision:** Esther Ariyo.

**Validation:** Esther Ariyo, Victoria Awortwe, Ebenezer Cudjoe.

**Writing – original draft:** Esther Ariyo.

**Writing – review & editing:** Esther Ariyo, Victoria Awortwe, Ebenezer Cudjoe.

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
