## [Decision Letter · Decision Letter 0]

16 Jul 2025

Dear Dr. Ariyo,

Thank you for submitting your manuscript to PLOS ONE. After careful consideration, we feel that it has merit but does not fully meet PLOS ONE’s publication criteria as it currently stands. Therefore, we invite you to submit a revised version of the manuscript that addresses the points raised during the review process.

We look forward to receiving your revised manuscript.

Kind regards,

Vidanka Vasilevski

Academic Editor

PLOS ONE

Journal Requirements: 

Additional Editor Comments:

Please address the comprehensive feedback provided by the reviewers.

Reviewers' comments:

Reviewer's Responses to Questions

**Comments to the Author**

1. Is the manuscript technically sound, and do the data support the conclusions?

Reviewer #1: Yes

Reviewer #2: Yes

Reviewer #3: Yes

2. Has the statistical analysis been performed appropriately and rigorously?

Reviewer #1: Yes

Reviewer #2: Yes

Reviewer #3: Yes

3. Have the authors made all data underlying the findings in their manuscript fully available?

Reviewer #1: Yes

Reviewer #2: Yes

Reviewer #3: Yes

4. Is the manuscript presented in an intelligible fashion and written in standard English?

Reviewer #1: No

Reviewer #2: Yes

Reviewer #3: Yes

Reviewer #1: This paper describes a systematic review of perinatal interventions for parents with histories of child removal, a significant topic that warrants further research. Overall, the topic is introduced well with relevant literature providing a strong rationale for the study. The methods of choice fit well with the stated research questions and overall appear to be conducted appropriately. Despite the significant findings from the review, the reporting of results appears inconsistent and hard to follow, leading to a weak discussion with clear conclusions and useful recommendations for future research and service design. Please see specific recommendations in the attached document, to improve the paper to a standard required for publication and ensure that the findings can offer the positive contribution to the literature that is warranted.

Reviewer #2: This manuscript is well put together and provides a balanced look at an important and under-researched topic: perinatal interventions for families at risk of having another child removed. It does a solid job of placing the issue in the context of current services and policy work, while also pointing out the big gaps in the existing evidence. The methods are robust and transparent, with a thorough search and use of well-established tools to assess study quality. The tables and narrative summaries help make sense of complex data, and the discussion strikes a good balance between careful interpretation and practical suggestions for policy and practice.

Minor revisions are suggested below:

PROSPERO registration

Page 8 (line 176 mentions “Registration number: XXXXXX”). Suggestion: Add the true registration number to complete the transparency of the protocol.

Parental acceptability

Page 32 (lines 555–580 in section 3.6). Suggestion: This section notes that direct parental perspectives were not captured, relying instead on inferred data. Explicitly stating this as a limitation in the conclusion or discussion would add important caution to this aspect of the paper.

Language clarity

Throughout the manuscript – but particularly noticeable in sections like the Abstract, Introduction and Results/Discussion. Suggestion: Minor grammatical tweaks and trimming of run-on sentences would enhance readability, especially for an international audience.

Reviewer #3: This paper would benefit from an overall review and revision to strengthen the written expression and structure throughout. Some minor grammatical errors or informalities are present throughout the manuscript, which require further proof reading and revision from the authors.

**Do you want your identity to be public for this peer review?** For information about this choice, including consent withdrawal, please see our Privacy Policy

Reviewer #1: No

Reviewer #2: No

Reviewer #3: No

---

## [Author Response · Author response to Decision Letter 1]

30 Aug 2025

Response to Reviewers

Reviewer Comments:

This is an interesting and meaningful systematic review which considers several elements of the effectiveness of interventions designed to prevent infant removals.

Thank you very much for your thoughtful and constructive feedback on the manuscript. We sincerely appreciate your comments regarding the introduction and methodology of the study.

This paper would benefit from an overall review and revision to strengthen the written expression and structure throughout. The authors may wish to review the appropriateness of information presented in each section, and if it is better suited to other sections within the paper, to avoid redundancy. Further feedback has been provided in relation to each section of the paper to help guide authors in their revision.

In light of your comments, the manuscript has been reviewed. The written structure has been addressed. For example. The result and discussion has been restructured and text about quality of Evidence is now put together as a subsection

Overarching Comments:

It is not clear throughout the paper if this review looks at interventions aimed at preventing recurrent infant removals (i.e., where parents have experienced previous infant removal) or infant removals in general. From the methods section and eligibility criteria, it appears to be the latter (interventions aimed at preventing infant removals in general, as there is no criteria based on previous removal experience).- If this is the case, the authors may consider removing emphasis on recurrent infant removals, as this confuses the intent and target population of the interventions.

The authors have removed the emphasis on recurrent infant removals and make it infant removal.

However, we do indicate that recurrent removal contributes significantly to the rate of incidence removal. This can be found in paragraph 2 of page 4 as

“Research shows that recurrent care proceedings account for a significant portion of infant removals across countries, with first-time mothers in England, Indigenous groups in Australia, and Black families in the USA identified as particularly affected populations (5-7). Recurrent care proceedings includes “a cycle of returning to court and having subsequent children removed”(8). An Australian cohort study of 1,629 mothers (2000–2018) found that 20.3% of biological parents who had an infant removed by court order experienced repeat removals, with Indigenous groups representing the largest proportion in national infant removal statistics(9, 10). In Scotland, Wales and England respectively at least twenty percent of mothers who experience infant removal , were first time mothers (11, 12).In England, about 25% of mothers with a prior court-ordered infant removal subsequently faced repeat care proceeding within 10years (13, 14). In England and Wales respectively, half of mothers involved in care proceedings were subject to repeat care proceedings within eight years and five years respectively (15). 92% of Scottish mothers with older children in care proceedings, had previously had at least one child removed(16). Studies in the U.S. show that birth mothers with a previous child removed have significantly higher odds of experiencing another removal (17), with black children over represented at 20% in 2021(18). “

The authors may consistently referring to 'infant removals' (as opposed to baby removals) throughout the paper.

This has been changed and there is a consistent use of the term infant removal in the text.

It could be individual style or preference, but I encourage the authors to be mindful of overusing hyphens in any discussion throughout the paper.-

This is noted and has been corrected

Introduction:

The introduction section would benefit from more detailed discussion on the impact of infant removals, specifically, (as opposed to child removal broadly), including long-term impacts on parents, infant etc. as infant removals present their own unique challenges and psychosocial, cultural, and socioeconomic impacts for families.

We thank the reviewer for highlighting this important point. We agree that infant removals have unique implications distinct from broader child removal and warrant specific discussion. In response, the Introduction has been tailored to include literature that are specific on the psychosocial, cultural, and socioeconomic impacts of infant removals, as well as long-term consequences for both infants and parents.

The introduction section would also benefit from some data demonstrating the rate of infant removals across the contexts of focus for the review, including data for Indigenous populations specifically, considering two of the included studies in the review focus on Indigenous mothers.

Thank you for this valuable suggestion. We agree that including data on the rate of infant removals across the contexts of focus would strengthen the introduction and provide important context. We have now incorporated relevant statistics highlighting infant removal rates, including data specific to Indigenous populations, to reflect the focus of the studies included in our review. This addition clarifies the scope and significance of the issue for the populations under study. It is as shown below

Emerging evidence shows a significant portion of child removals involve infants (1). In Australia, infants (children under 1 year old) represent the largest proportion of child removals, with a removal rate of 6.5 percent 1,000 which is significantly higher than the rate of 2.0 per 1,000 observed in other age groups between 2023 and 2024(2). Similarly in England, 70 per 10000 children under one year old were in care proceeding in comparison to 19 per 10,000 children aged between one and four years old(3). It can be found on Page 4

Paragraph 2, Page 5 - This paragraph could be strengthened with a more detailed overview of early intervention service delivery (e.g., what types of support are being provided) and the elements which are thought to be important in effective early intervention strategies (e.g., see Keddell et al., 2023).

This has been addressed and the text now includes a short description of early service intervention as shown below and can be found on page 6

“There is growing but still limited evidence that intensive, relational, and tailored interventions, that combines practical, therapeutic, emotional support and advocacy, with a focus on reducing parental stress and drawing on parents’ commitment to their children can be effective in reducing the rate of infant removal”

Objectives section - Rationales for incl./excl. criteria to be moved to eligibility criteria section. Suggest moving the definitions and descriptions of recurrent care interventions and care proceedings into the introduction, where these concepts can be introduced to the reader appropriately.

The definitions has been moved to the introduction first paragraph and rationales have been moved to the eligibity criteria section

Methodology:

Paragraph 2, Page 8 - 'We conducted a systematic review to help us understand the interventions that prevent...' - suggest rewording 'us' to the wider academic audience - (e.g., it is also for the field to obtain a better understanding).

We have now reworded the sentence. Please see page 9, lines 183-185: We conducted this systematic review to investigate the effectiveness of perinatal interventions designed to prevent infant removals and recurrent care proceedings, including the context in which they work, focusing on interventions that begin during pregnancy.

'A meta-analysis was not deemed possible based on preliminary literature search and evidence from literature' - While I appreciate grounds for not conducting a meta-analysis for this synthesis of literature, could the authors clarify this justification further for the reader (e.g., evidence was too heterogeneous?).

We have now strengthened the justification for not conducting meta-analysis. Please see page 13, lines 289-290: We did not conduct a meta-analysis given the substantial heterogeneity in the results of individual studies which could undermine the validity of the pooled results (36).

Paragraph 1, Page 10 - Could the authors briefly describe the child protection system orientation which is shared across the contexts mentioned (in relation to criteria ii).

A description of the chid protection orientation and similarity across the context is included in the criteria. It is now numbered as criteria iii

“These countries share a child protection orientation that is broadly comparable to the UK model, in which the state holds statutory powers to intervene in cases of child maltreatment, including the authority to remove children from parental care through court proceedings. These systems are primarily child-focused, operate under a legal framework that prioritises the child’s safety and welfare, and allow for both voluntary and involuntary out-of-home placements. They also share features such as adversarial or quasi-adversarial court processes, mandatory reporting duties, and a focus on preventing re-entry into care, making them relevant for comparison with the UK context of high rates of infant removal and repeat removals” page

Paragraph 1, Page 10 - Minor correction - these are three distinct criteria (in relation to criteria iii).

The criteria has been split into 2

Paragraph 1, Page 10 – As per earlier comment, there is no mention in the eligibility criteria of parents previous removal history (i.e., if interventions were targeted towards parents who had a previous experience of infant/child removal). As this paper aims to investigate the effectiveness of perinatal interventions designed to prevent recurrent infant removals, it is unclear if the interventions reviewed are effective at preventing recurrent removals, or removals in general. Please clarify this in the eligibility criteria and how it was ensured interventions were targeted towards preventing recurrent infant removals, specifically.

The focus of the intervention is not on recurrent care only, but on general infant removal. We have included a statement about this and mentioned how the interventions are targeted towards prevent infant removal. The criteria has specified this.

“It includes an intervention aimed at preventing child removal for pregnant mothers at risk of child removal. “ page 11

The relevance of recurrent care removal to general infant removal is discussed in the introduction.

Paragraph 1, Page 11 - Could the authors comment on or provide the result of interrater consistency/agreement?

We have comment about interrater agreement as “Interrater agreement during screening was moderate, with a proportionate agreement of 71% and a Cohen’s kappa of 0.39, indicating fair agreement (47).” Page 12

Paragraph 2, Page 11 - Could the authors briefly describe the difference between descriptive and thematic analysis, if distinguishing between the two for different research questions?

We thank the reviewer for this helpful comment. We agree that it is important to clarify the distinction between descriptive and thematic analysis in relation to our research questions.The manuscript now includes

“For research question one, which focused on identifying interventions and policies that reduce infant removals, we used descriptive analysis to summarise and categorise the types of interventions reported across studies. This approach allowed us to map the range, and characteristics of interventions in a structured way. For research questions two to five, which explored effectiveness across different parent groups, common elements of interventions, implementation factors, and parents’ perspectives, we used thematic analysis. (46). This approach enabled us to synthesise qualitative and mixed-methods findings, identify recurring patterns, and generate overarching themes across studies. We then provide a narrative of all findings in alignment with our study objectives. Data was synthesised per each research objective. We did not conduct a meta-analysis given the substantial heterogeneity in the results of individual studies which could undermine the validity of the pooled results”page 13

Results:

Paragraph 1, Page 13 - Could the authors clarify how at risk of child removal was defined in these studies (e.g., received one or more pre-birth CP notifications, or, previous history of child removal?)

We have included how risk of infant removal was defined in the studies and is as shown below

“All studies included mothers that were at risk of child removal based on previous child removal (53, 54), or referral to child protection service (30, 52), vulnerable Indigenous (first nations) mothers (50, 51)” page 15, line 324-326

Paragraph 2, Page 13 - Could the authors describe what they're referring to as 'program data' here?

This is now included and stated as

“Data sources included program data collected during the implementation of the intervention (50, 53), administrative datasets (30, 51) and a combination of program data and validated measures (31, 52).” Page 15, line 327

Paragraph 1, Page 14 - It would be beneficial to see further description here of the policy reform intervention, and what exactly this entailed/how it was implemented.

We appreciate this suggestion and have now provided a clearer description of the prenatal reporting policy and its implementation.It is included as shown below

“The prenatal reporting policy includes the notification of child protection services during pregnancy about an unborn child that is considered at risk, with the aim of preventing harm after birth or having the baby taken into care and it offered voluntary assessments, child protection assessment at birth, referral to support services, and case work support for the coordination of services (30)” page 16 line 345

Paragraph 2, Page 14 - Quick correction: interventions*

Thanks for the observation, this is corrected

Paragraph 2, Page 15 - The finding in relation to reference (39), demonstrating those in the intervention group to have increased rates of infant removal compared to the control group is very interesting. I would be interested in commentary or some insight to this in the discussion section.

We have include a commentary about this into the manuscript

“The authors suggests that this likely reflects a more timely and informed decision making by practitioners, and action to prevent longer term harm. ” page 18,line 386

Paragraph 1, Page 23 – Indigenous needs a capital ‘I’

Thanks , this is corrected

It would be great if the authors included some further descriptions of the culturally safe support strategies implemented in some of the interventions and found to have a positive effective/be an enabler.

We thank the reviewer for this suggestion. We have now included additional details of the culturally safe strategies within the BiOC model, highlighting Indigenous leadership and governance, continuity of care through midwives and First Nations Family Support Workers, staff training in cultural integrity frameworks, and community hubs that promote connection, support, and engagement with Elders as shown below

“BiOC’s multi-agency partnership model, led by Indigenous organisations, successfully reduced newborn removals through culturally safe, multi sectoral support to women. Key characteristics included Indigenous leadership and governance, continuity of care provided by dedicated midwives and Indigenous family support workers, integration of cultural integrity frameworks into staff training, and the creation of community hubs that fostered connection, peer support, and engagement with indigenous elders(51)”page 28 line 533

Discussion:

Although I appreciate the reviews’ focus of intervention effectiveness at preventing infant removals through supporting parents in minimising child protection risks, it would be good to see some acknowledgement in the discussion of the issue of heightened surveillance and bias in child protection practices and interventions, which can often discriminate against and are inequitable for certain minority or racial groups.

This now included in the Conclusion and impication and written as shown below

---

## [Decision Letter · Decision Letter 1]

13 Oct 2025

Dear Dr. Ariyo,

Thank you for submitting your manuscript to PLOS ONE. After careful consideration, we feel that it has merit but does not fully meet PLOS ONE’s publication criteria as it currently stands. Therefore, we invite you to submit a revised version of the manuscript that addresses the points raised during the review process.

We look forward to receiving your revised manuscript.

Kind regards,

Vidanka Vasilevski

Academic Editor

PLOS ONE

Journal Requirements:

Reviewers' comments:

Reviewer's Responses to Questions

**Comments to the Author**

Reviewer #1: All comments have been addressed

Reviewer #2: All comments have been addressed

Reviewer #3: All comments have been addressed

2. Is the manuscript technically sound, and do the data support the conclusions?

Reviewer #1: Yes

Reviewer #2: Yes

Reviewer #3: Partly

3. Has the statistical analysis been performed appropriately and rigorously?

Reviewer #1: Yes

Reviewer #2: Yes

Reviewer #3: N/A

4. Have the authors made all data underlying the findings in their manuscript fully available?

Reviewer #1: Yes

Reviewer #2: Yes

Reviewer #3: Yes

5. Is the manuscript presented in an intelligible fashion and written in standard English?

Reviewer #1: No

Reviewer #2: Yes

Reviewer #3: Yes

Reviewer #1: Thank-you for addressing all the queries to your initial submission. I believe the manuscript has improved greatly and offers a significant and meaningful contribution to the literature. With the significant revisions made, there are many language, formatting and grammatical errors to correct. Please review the manuscript carefully for the following suggestions.

1. Consistency in your use of USA or U.S/UK and United Kingdon, being sure to spell all acronyms out when first using them in the manuscript i.e. Page 4 Line 83 / Page 5 Line 94

2. Consistency in spelling out the word for numbers ten and below, and using numerical figures for numbers greater than 10 e.g. Page 4 Line 73 and many instances throughout the manuscript

3. Page 4 - Check for consistency in capitalization for the word 'Black'

4. Page 4 Line 93 - Avoid starting the sentence with a numerical figure, unless writing it out full in text.

5. Page 9 Line 191 - Please capitalise APA PsycInfo and PsycArticles correctly and ensure this is consistent with Figure 1

6. Page 12 Line 266 - be consistent with capitalising (or not) the first word in each dot point of the exclusion criteria

7. Page 15 Line 320 - Please edit to read "None of the included studies reported on any interventions or engagement with fathers"

8. Page 15 Line 326 - Please capitalise First Nations

9. Please spell out and capitalise proper nouns correctly, e.g. Page 15, Line 337 Australian Capital Territory; Page 16 Line 342 Parents under Pressure

10. Page 16 Line 358 - Please reword 'till' to 'until' or 'to'

11. Page 17 Line 365 - Please correct the grammar in the final sentence and change question to 'questions'

12. Please check all formatting is correct in Table 1, including consistent use of spaces between statistical results e.g. = % <; language is clear and grammatically correct e.g. Reference (32) child removal outcome

13. Page 17 and other places in manuscript - be consistent with terminology infants rather than babies as previous noted

14. Please consider the overall formatting of Tables 2 and 3 and whether the columns could instead be rows and the rows be columns, to fit more clearly within the portrait frame

15. Page 23 Line 420 - Please correct the word 'present' to 'presents'

16. Page 26 Line 292 - Please write 'approaches' in the heading and first sentence. Please further check this paragraph for correct formatting

17. Please capitalise the word Table on Page 27 Line 519

18. Please include all studies in Table 4 or consider the usefulness of the table outside of describing the four common elements and four studies that reported on them.

19. Page 28 - Consider being more concise with your subheadings in Section 3.6. and removing the words 'As a barrier"

20. Page 29 - please check formatting on this page and consistency of intext referencing

21. Page 29 - consider rewording the title of subheading 3.6.2. to "Fear of child removal and distrust of services"

22. Page 29 Line 566-567 - Please improve the clarity and accuracy of this sentence using the word mothers twice

23. Page 31 Section 3.2 - Please review and enhance the clarity of this first paragraph.

24. Page 32 Line 616 - please change the word 'staffs'

25. Page 32 Lines 626-627 - Please review and improve the wording of this statement to be clearer and accurate. E.g. While further research is needed on parental perspectives, for those studies that did address acceptability and engagement of interventions, reports were high (as displayed in Table 3).

26. Please review wording and formatting in Table 6 and improve readability

27. Page 36 Line 666 - Remove the word 'the' before 'infant'

28. Page 37 - Please revise and improve the language in this first paragraph. The last sentence is very long and difficult to read.

29. Page 37 Line 677 - Please clarify "intervention effectiveness in reducing infant removals"

30. Page 37 Line 685 - Consider clarifying the "need for greater quality and strength of evidence on ...." and improving the wording of the following sentence cross Lines 686 - 689

31. Page 38 Line 697 - Please consider why only the UK as the review has also highlighted need related to particular groups in other countries i.e. First Nations parents in Australia

Reviewer #2: The authors have responded to all feedback and this article is ready for publication. They are to be congratulated.

Reviewer #3: (No Response)

**Do you want your identity to be public for this peer review?** For information about this choice, including consent withdrawal, please see our Privacy Policy

Reviewer #1: No

Reviewer #2: No

Reviewer #3: No

---

## [Author Response · Author response to Decision Letter 2]

4 Nov 2025

Response to Reviewers

Reviewer Comments:

1. Consistency in your use of USA or U.S/UK and United Kingdon, being sure to spell all acronyms out when first using them in the manuscript i.e. Page 4 Line 83 / Page 5 Line 94

Response: Corrected to “and Black families in the United States of America (USA) identified-

proceedings had previously had at least one child removed (16). Studies in the USA show ”

2. Consistency in spelling out the word for numbers ten and below, and using numerical figures for numbers greater than 10 e.g. Page 4 Line 73 and many instances throughout the manuscript

Response: Corrected to “Australia, infants (children under one year old) represent the largest proportion of child”

3. Page 4 - Check for consistency in capitalization for the word 'Black'

All changed

4. Page 4 Line 93 - Avoid starting the sentence with a numerical figure, unless writing it out full in text.

Response: Corrected to “Ninety two percent of Scottish mothers with older…”

5. Page 9 Line 191 - Please capitalise APA PsycInfo and PsycArticles correctly and ensure this is consistent with Figure 1

Response: Corrected to “We systematically searched six electronic databases (i.e., APA PsycINFO, PsycARTICLES, Scopus, PubMed, CINAHL, and Web of Science).

6. Page 12 Line 266 - be consistent with capitalising (or not) the first word in each dot point of the exclusion criteria

Response: Corrected to “were abstracts, conference proceedings, reviews and grey literature.”

7. Page 15 Line 320 - Please edit to read "None of the included studies reported on any interventions or engagement with fathers"

Response: Corrected to “None of the included studies reported on any intervention with fathers or engagement with fathers. “

8. Page 15 Line 326 - Please capitalise First Nations

Response: Corrected to “vulnerable Indigenous (First Nations) mothers”

9. Please spell out and capitalise proper nouns correctly, e.g. Page 15, Line 337 Australian Capital Territory; Page 16 Line 342 Parents under Pressure

Done

10. Page 16 Line 358 - Please reword 'till' to 'until' or 'to'

Response: Corrected to “Three interventions provided support up untill one year after birth

11. Page 17 Line 365 - Please correct the grammar in the final sentence and change question to 'questions'

Done

12. Please check all formatting is correct in Table 1, including consistent use of spaces between statistical results e.g. = % <; language is clear and grammatically correct e.g. Reference (32) child removal outcome

Response: Table is now corrected

13. Page 17 and other places in manuscript - be consistent with terminology infants rather than babies as previous noted

Response: All changed to infants

14. Please consider the overall formatting of Tables 2 and 3 and whether the columns could instead be rows and the rows be columns, to fit more clearly within the portrait frame

Response: Table is now corrected

15. Page 23 Line 420 - Please correct the word 'present' to 'presents'

Done

16. Page 26 Line 292 - Please write 'approaches' in the heading and first sentence. Please further check this paragraph for correct formatting

Done

17. Please capitalise the word Table on Page 27 Line 519

Done

18. Please include all studies in Table 4 or consider the usefulness of the table outside of describing the four common elements and four studies that reported on them.

Response: Table is now corrected to include all studies

19. Page 28 - Consider being more concise with your subheadings in Section 3.6. and removing the words 'As a barrier"

Response: 'As a barrier “is removed

20. Page 29 - please check formatting on this page and consistency of intext referencing

Response: formatting on page is now corrected

21. Page 29 - consider rewording the title of subheading 3.6.2. to "Fear of child removal and distrust of services"

Done

22. Page 29 Line 566-567 - Please improve the clarity and accuracy of this sentence using the word mothers twice

Done

23. Page 31 Section 3.2 - Please review and enhance the clarity of this first paragraph.

Done

24. Page 32 Line 616 - please change the word 'staffs'

Done

25. Page 32 Lines 626-627 - Please review and improve the wording of this statement to be clearer and accurate. E.g. While further research is needed on parental perspectives, for those studies that did address acceptability and engagement of interventions, reports were high (as displayed in Table 3).

Done

26. Please review wording and formatting in Table 6 and improve readability

Done

27. Page 36 Line 666 - Remove the word 'the' before 'infant'

Done

28. Page 37 - Please revise and improve the language in this first paragraph. The last sentence is very long and difficult to read.

Done

29. Page 37 Line 677 - Please clarify "intervention effectiveness in reducing infant removals"

Done

30. Page 37 Line 685 - Consider clarifying the "need for greater quality and strength of evidence on ...." and improving the wording of the following sentence cross Lines 686 – 689

Done. Changed into “Although there are evaluation reports that address recurrent care proceedings and infant removal, only six peer-reviewed studies met the inclusion criteria for this review”

31. Page 38 Line 697 - Please consider why only the UK as the review has also highlighted need related to particular groups in other countries i.e. First Nations parents in Australia

Changed to “and especially in the context where rates of recurrent care proceedings still persist”

---

## [Editor Report · Decision Letter 2]

13 Nov 2025

Breaking the Cycle: Systematic Review of Perinatal Interventions for Parents at Risk of Child Removal

PONE-D-25-19264R2

Dear Dr. Ariyo,

We’re pleased to inform you that your manuscript has been judged scientifically suitable for publication and will be formally accepted for publication once it meets all outstanding technical requirements.

Kind regards,

Vidanka Vasilevski

Academic Editor

PLOS ONE

---

## [Editor Report · Acceptance letter]

PONE-D-25-19264R2

PLOS ONE

Dear Dr. Ariyo,

I'm pleased to inform you that your manuscript has been deemed suitable for publication in PLOS ONE. Congratulations! Your manuscript is now being handed over to our production team.

Kind regards,

on behalf of

Dr. Vidanka Vasilevski

Academic Editor

PLOS ONE